# Protective Effects of a New C-Jun N-terminal Kinase Inhibitor in the Model of Global Cerebral Ischemia in Rats

**DOI:** 10.3390/molecules24091722

**Published:** 2019-05-03

**Authors:** Mark B. Plotnikov, Galina A. Chernysheva, Oleg I. Aliev, Vera I. Smol’iakova, Tatiana I. Fomina, Anton N. Osipenko, Victoria S. Rydchenko, Yana J. Anfinogenova, Andrei I. Khlebnikov, Igor A. Schepetkin, Dmitriy N. Atochin

**Affiliations:** 1Department of Pharmacology, Goldberg Research Institute of Pharmacology and Regenerative Medicine, Tomsk NRMC, Tomsk 634028, Russia; mbp2001@mail.ru (M.B.P.); bona2711@mail.ru (G.A.C.); oal67@yandex.ru (O.I.A.); light061265@mail.ru (V.I.S.); 2National Research Tomsk State University, Tomsk 634050, Russia; 3Department of Medicine Toxicology, Goldberg Research Institute of Pharmacology and Regenerative Medicine, Tomsk NRMC, Tomsk 634028, Russia; toxicology_lab@mail.ru; 4Department of Pharmacology, Siberian State Medical University, Tomsk 634050, Russia; osipenko-an@mail.ru; 5Department of Biophysics, Siberian State Medical University, Tomsk 634050, Russia; ryd4enkoviknoriya@mail.ru; 6Cardiology Research Institute, Tomsk NRMC, Tomsk 634012, Russia; cardio.intl@gmail.com; 7Kizhner Research Center, Tomsk Polytechnic University, Tomsk 634050, Russia; aikhl@chem.org.ru (A.I.K.); igor@montana.edu (I.A.S.); 8Research Institute of Biological Medicine, Altai State University, Barnaul 656049, Russia; 9Department of Microbiology and Immunology, Montana State University, Bozeman, MT 59717, USA; 10Cardiovascular Research Center, Cardiology Division, Massachusetts General Hospital, Harvard Medical School, Charlestown, MA 02129, USA

**Keywords:** c-Jun N-terminal kinase, JNK inhibitor, neuroprotection, model of global cerebral ischemia, antiradical activity, cerebral microcirculation

## Abstract

c-Jun N-terminal kinase (JNK) is activated by various brain insults and is implicated in neuronal injury triggered by reperfusion-induced oxidative stress. Some JNK inhibitors demonstrated neuroprotective potential in various models, including cerebral ischemia/reperfusion injury. The objective of the present work was to study the neuroprotective activity of a new specific JNK inhibitor, IQ-1S (11*H*-indeno[1,2-*b*]quinoxalin-11-one oxime sodium salt), in the model of global cerebral ischemia (GCI) in rats compared with citicoline (cytidine-5′-diphosphocholine), a drug approved for the treatment of acute ischemic stroke and to search for pleiotropic mechanisms of neuroprotective effects of IQ-1S. The experiments were performed in a rat model of ischemic stroke with three-vessel occlusion (model of 3VO) affecting the brachiocephalic artery, the left subclavian artery, and the left common carotid artery. After 7-min episode of GCI in rats, 25% of animals died, whereas survived animals had severe neurological deficit at days 1, 3, and 5 after GCI. At day 5 after GCI, we observing massive loss of pyramidal neurons in the hippocampal CA1 area, increase in lipid peroxidation products in the brain tissue, and decrease in local cerebral blood flow (LCBF) in the parietal cortex. Moreover, blood hyperviscosity syndrome and endothelial dysfunction were found after GCI. Administration of IQ-1S (intragastrically at a dose 50 mg/kg daily for 5 days) was associated with neuroprotective effect comparable with the effect of citicoline (intraperitoneal at a dose of 500 mg/kg, daily for 5 days).The neuroprotective effect was accompanied by a decrease in the number of animals with severe neurological deficit, an increase in the number of animals with moderate degree of neurological deficit compared with control GCI group, and an increase in the number of unaltered neurons in the hippocampal CA1 area along with a significant decrease in the number of neurons with irreversible morphological damage. In rats with IQ-1S administration, the LCBF was significantly higher (by 60%) compared with that in the GCI control. Treatment with IQ-1S also decreases blood viscosity and endothelial dysfunction. A concentration-dependent decrease (IC_50_ = 0.8 ± 0.3 μM) of tone in isolated carotid arterial rings constricted with phenylephrine was observed after IQ-1S application in vitro. We also found that IQ-1S decreased the intensity of the lipid peroxidation in the brain tissue in rats with GCI. 2.2-Diphenyl-1-picrylhydrazyl scavenging for IQ-1S in acetonitrile and acetone exceeded the corresponding values for ionol, a known antioxidant. Overall, these results suggest that the neuroprotective properties of IQ-1S may be mediated by improvement of cerebral microcirculation due to the enhanced vasorelaxation, beneficial effects on blood viscosity, attenuation of the endothelial dysfunction, and antioxidant/antiradical IQ-1S activity.

## 1. Introduction

c-Jun N-terminal kinase (JNK) is a critical mitogen activated protein kinase (MAPK) modulated by various brain stimuli and implicated in neuronal injury triggered by reperfusion-induced oxidative stress [1]. Three distinct JNKs, designated as JNK1, JNK2, and JNK3, have been identified, and at least 10 different splicing isoforms exist in the mammalian cells [2]. JNK3 is found almost exclusively in the brain [3] but it is not dominant, as JNK3 knockout results only in a weak attenuation of the total JNK pool in the brain tissue [4]. Increased JNK phosphorylation and JNK activity in the hippocampus have been reported after cerebral ischemia and reperfusion injury [5,6]. Sustained JNK activation has been shown to be associated with neuronal death and apoptosis following ischemic stroke. Various nonprotein synthetic inhibitors of JNK enzymatic activity are described, including SP600125, AS601245, IQ-1S, SR-3306, and SU3327 as well as protein and nonprotein molecules inhibiting interactions of JNKs with their substrates, folding proteins, and/or cell organelles [7,8,9,10,11,12,13]. Some of them demonstrated neuroprotective activity in the animal models of stroke [14,15,16,17]. The specific JNK inhibitors selectively attenuate the activity of brain-expressed JNK3 and/or JNK signaling pathway that form in pathological conditions of ischemia/reperfusion [13,18]. These data suggest that searching for the efficacious neuroprotectors among JNK inhibitors is a promising area for the development of approaches to treatment of acute ischemic and reperfusion brain injury [13,18].

A new specific JNK inhibitor, IQ-1S (11*H*-indeno[1,2-*b*]quinoxalin-11-one oxime sodium salt), has high affinity to JNK3 compared with its affinity to JNK1/JNK2 [9,19]. Evaluation of the therapeutic potential of IQ-1S showed that the compound could inhibit matrix metalloproteinase 1/3 gene expression in human synoviocytes and significantly attenuated development of murine collagen-induced arthritis [19]. IQ-1S also suppressed lipopolysaccharide-induced production of proinflammatory cytokines in a culture of monocytic cells [9]. Moreover, IQ-1S could release nitric oxide (NO) during its enzymatic metabolism by liver microsomes and serum nitrite/nitrate concentration in mice after intraperitoneal (i.p.) injection of IQ-1S was increased [20]. It should be noted that JNK signaling pathway is closely associated with NO production in ischemia and reperfusion [21]. Donors of exogenous NO decrease S-nitrosylation of mixed-lineage protein kinase 3 (MLK3) triggered by reperfusion and inhibit activation of JNK-dependent pathway [22]. A NO donor, sodium nitroprusside (SNP), decreases JNK3 phosphorylation and damage to the hippocampal neurons after global ischemia/reperfusion [23]. Due to the dual action as a JNK inhibitor and a NO-donor, IQ-1S may have therapeutic potential, which was previously evaluated in an animal stroke model. In the model of focal cerebral ischemia in mice, IQ-1S demonstrated remarkable reductions in neurological deficit and infarct volume as compared with vehicle-treated mice after 2 days of reperfusion [20].

Updated Stroke Treatment Academic Industry Roundtable (STAIR) recommendations set a variety of requirements for research of promising neuroprotectors [24]. One of these requirements demands that efficacy of a potential neuroprotector must be established in at least two species using both histological and behavioral outcome measures. The aim of the present work was to study the neuroprotective activity and the mechanisms of IQ-1S action in the model of transient global cerebral ischemia (GCI) in rats. We also evaluated the content of lipid peroxidation products in the brain tissue, local cerebral blood flow (LCBF), blood viscosity, hematocrit, parameters of erythrocyte aggregation, erythrocyte elongation index, indices of plasma hemostasis, and vasodilator activity of endothelium in IQ-1S-treated and nontreated rats at day 5 after GCI. Because previous data demonstrated that JNK contributes to the regulation of vascular function, including vascular smooth muscle contraction [25], an effect of IQ-1S on the vascular tone of the isolated carotid artery rings in vitro was also studied.

## 2. Results

### 2.1. Effects of IQ-1S on Survival and Neurological Status

In group of sham-operated animals, spontaneous motor activity decreased in 38% and 13% of animals at days 1 and 3 after surgical intervention, respectively; mean scores of neurological deficit were 0.4 ± 0.2 and 0.1 ± 0.1 for these time points, respectively. To day 5 of the study, no further changes in neurological status were found (Figure 1A).

After GCI reproduction in 45 rats of the control group for 5 days, 10 rats died (mortality rate was 22%). In experimental group administered with IQ-1S, seven (15%) rats of 46 animals died after ischemia episode. In the group of animals with i.p. introduction of citicoline after GCI, two rats (10%) of 20 animals died. No significant differences in the levels of mortality between groups were found (Table 1).

In control animals with GCI, neurological symptoms in the early reperfusion period (for the first 3 h) were characterized by the predominance of cerebral symptoms: disruption of spontaneous breathing, failure to maintain posture, areflexia, and hypertonia of the extremities. In surviving animals, the functions of the central neural system gradually restored and the neurological abnormalities regressed during the first hours after GCI. However, abnormalities in the muscle tone and motor coordination, slow or absent reflexes, and ptosis persisted up to day 5; some animals presented with convulsive disorder. At day 1 after reperfusion, mean score of neurological deficit was 8.4 ± 0.9. To days 3 and 5, mean scores of neurological deficit decreased to 82% and 61%, respectively, relative to the corresponding value at day 1; during the entire period of observation, significant differences persisted compared with the group of sham-operated animals (Figure 1A). At day 1 after GCI, all survived rats of control group had severe neurological deficit. At day 3 and day 5, neurological disorders persisted (Figure 1B).

In rats administered with IQ-1S at day 1 after GCI, mean score of neurological deficit was 6.1 ± 0.5 and was by 27% lower than the corresponding value in the control group. To days 3 and 5, mean scores of neurological deficit decreased to 4.2 ± 0.4 and 3.0 ± 0.3 suggesting that this indicator was significantly lower than control values by 39% and 41%, respectively (Figure 1A). In response to IQ-1S, at day 1 after GCI, the number of animals with severe neurological deficit significantly decreased (Figure 1B).

In the group of animals administered with citicoline, a drug approved for the treatment of acute ischemic stroke, mean score of neurological deficit at day 1 after reperfusion was 5.5 ± 0.8, which was 34% lower than the control value. To days 3 and 5, mean scores of neurological abnormalities decreased by 22% and 40% compared with the corresponding value at day 1 and were lower than the control values by 38% and 35%, respectively. During all periods of the study, the indicator significantly differed from the corresponding value in the group of control animals (Figure 1A). In response to citicoline at day 1 after reperfusion, the number of animals with severe neurological deficit significantly decreased (Figure 1B).

### 2.2. Effects of IQ-1S on the Morphological Structure of the CA1 Hippocampal Area

In sham-operated animals, the neurons of the CA1 hippocampal area had a round shape and homogenous staining of the cytoplasm and the nucleus, suggesting the absence of pathomorphological alterations. Numerical density of the neurons was 2,875 ± 121 cells per mm^2^ (Figure 2A,B). Modeling of GCI led to massive loss of the neurons in the pyramid layer of CA1 hippocampal region at day 5 after reperfusion. This manifested as a reduction of numerical neurons by 29% compared with sham-operated animals. Among surviving neurons, 42% of the cells were piknotic. Besides, we observed the layer disorganization; 6% of the cells had rim of pericellular edema of the neurons, and many neurons had eosinophilic cytoplasm (Figure 2A–D).

Comparison of the cell composition and neuronal morphology in the hippocampal CA1 region in the control group and in the experimental group with IQ-1S administration showed an increase in the numerical density of the neurons by 39% and over two-fold increase of unaltered neurons due to a decrease in the number of neurons with pathomorphological changes. Indeed, the number of neurons with pericellular edema in rats treated with IQ-1S was by 60% lower than the corresponding value in the control group (Figure 2C,D). Citicoline did not affect the number of neurons with pericellular edema (Figure 2C).

Therefore, IQ-1S, when administered to animals intragastrically (i.g.) at a dose of 50 mg/kg, exerted a pronounced neuroprotective effect seen as significant increases in the numerical density of the neurons, the number of vital pyramidal neurons, and a decrease in the number of neurons with pericellular edema in the hippocampal CA1 region. The extents of neuroprotective effects of IQ-1S at a dose of 50 mg/kg and citicoline at a dose of 500 mg/kg were comparable.

### 2.3. Effects of IQ-1S on Lipid Peroxidation in Cerebral Tissue

Lipid peroxidation products (LPPs) can serve as second messengers that evoke proapoptotic responses via covalent modification of the key sensors in neuronal tissue [26]. Therefore, as the next step of experiments, we evaluated the LPPs in the brain tissue of experimental animals after GCI. In control group at day 5 after GCI, we found a significant increase in diene conjugates and triene conjugates by 22% and 30%, respectively, compared with the corresponding values in sham-operated animals (Figure 3). IQ-1S (50 mg/kg, i.g.) decreased the intensity of the LPP formation in the brain tissue; the levels of diene conjugates and triene conjugates were significantly lower than in control by 16% and 25%, respectively (Figure 3). Citicoline (500 mg/kg, i.p.) also caused decreases in the levels of diene conjugates and triene conjugates (by 26% and 32%, respectively) in the brain tissue (Figure 3).

### 2.4. Antiradical Effect of IQ-1S

The values of specific reaction rates of the studied compounds with 2.2-diphenyl-1-picrylhydrazyl (DPPH) are presented in Figure 4 and Table 2. DPPH radical was stable in all used solvents. During 1-h registration in the presence of DPPH alone, no more than 0.5% of DPPH underwent transition to the nonradical form. Analysis of obtained data allowed to find that the specific reaction rate for the DPPH scavenging in ethyl acetate for IQ-1S did not significantly differ from ionol, whereas, in acetonitrile and acetone, it significantly exceeded the corresponding values for this control antioxidant (Figure 4A–C). On the contrary, specific reaction rate for DPPH scavenging in proton solvent ethanol for ionol was significantly higher than for IQ-1S (Figure 4D).

### 2.5. Effects of IQ-1S on LCBF in Cerebral Cortex, Mean Arterial Blood Pressure, and Heart Rate

The level of LCBF in intact rats was 1900 ± 191 blood perfusion unit (BPU). In the control group at day 5 after GCI, LCBF decreased by almost two times compared with sham-operated animals. In rats of the experimental group after IQ-1S administration (50 mg/kg, i.g.), LCBF was significantly higher (by 60%) compared with that in the control. No significant differences in the values of mean arterial blood pressure and heart rate after GCI were found between groups at day 5 (Table 3).

### 2.6. Effects of IQ-1S on Hemorheological Parameters

In the control group of animals at day 5 after creation of GCI model, we observed a significant increase in blood viscosity in the entire diapason of shear rates (by 14–16%) compared with sham-operated rats (Figure 5). In animals of the control group, we found a significant increase in plasma viscosity (by 3%) and hematocrit (by 7%) relative to the corresponded values in the group of sham-operated animals (Table 4). Administration of IQ-1S (50 mg/kg, i.g.) to rats with GCI limited an increase in whole blood viscosity in a wide range of shear rates (from 60 to 450 s^−1^). Blood viscosity was lower than that in control by 9–11% (Figure 5). Plasma viscosity and hematocrit values in animals with i.g. administration of IQ-1S were preserved at the levels of those in sham-operated animals (Table 4).

The study of erythrocyte aggregation in shear flow in animals with GCI did not show any significant differences between sham-operated rats, control rats, and animals administered with IQ-1S (Table 4). The study of erythrocyte deformability in animals with GCI showed a decrease of elongation index by 2–4% in the presence of shear stress of 1–3 Pa in animals of the control group compared with that in sham-operated animals, but these changes did not reach statistical significance. In group of animals administered with IQ-1S, no changes in erythrocyte elongation index were found (Table 5).

### 2.7. Effects of IQ-1S on Vasodilator Activity of Endothelium

Evaluation of the endothelial function in sham-operated rats showed that a brief decrease in systolic blood pressure, followed by its rapid recovery, occurred in response to intravenous (i.v.) bolus introduction of acetylcholine (Ach) at a dose of 40 µg/kg (area above the curve of blood pressure decrease, S_Ach_, was 853 ± 153 mm Hg•sec). In response to i.v. bolus administration of SNP at a dose of 30 µg/kg, similar response of arterial blood pressure occurred (area above the curve of blood pressure decrease, S_SNP_, was 1900 ± 282 mm Hg•sec). The index of vasodilatory activity (IVA) calculated as ratio of S_Ach_/S_SNP_ in sham-operated animals was 0.41 ± 0.04 (Figure 6).

In rats with GCI, endothelium-dependent reduction in systolic blood pressure in response to acetylcholine was lower. Indeed, there was a significant 29% decrease in S_Ach_ compared with the corresponding values in group of sham-operated rats. In the absence of changes in S_SNP_, IVA decreased, which suggested the development of endothelial dysfunction in rats after GCI (Figure 6).

In group of GCI rats administered with IQ-1S at a dose of 50 mg/kg, the value of S_Ach_ was significantly higher than that in the group of control animals, which was seen as expected increase in IVA by 25% compared with that in the control (Figure 6). Citicoline in these conditions did not affect the study parameters of endothelial function in rats after GCI.

### 2.8. Effects of IQ-1S on Tonus of Isolated Carotid Artery Rings

In our study, no significant differences between the mean values of arterial blood pressure were observed in the experimental groups (Table 3). These findings suggest that the changes in the perfusion pressure do not modulate the effects of IQ-1S on LCBF. In search for other mechanisms of LCBF increase, the effects of IQ-1S on the vascular tone of carotid arteries in vitro were studied. When IQ-1S was applied at concentrations ranging from 0.001 to 100 μM to carotid arterial rings constricted with phenylephrine, concentration-dependent decrease (IC_50_ = 0.8 ± 0.3 μM) in the vascular smooth muscle ring tone was observed (Figure 7).

### 2.9. Effects of IQ-1S on Indices of Plasma Hemostasis

At day 5 after GCI, a clear tendency to increasing fibrinogen level (Fbg) and shortening thrombin time (TT) was observed in control group relative to sham-operated animals (Table 6). Parameters of activated partial thromboplastin time (aPTT) and prothrombin time (PT) did not differ from the corresponding values in sham-operated animals. In the presence of course intragastric administration of IQ-1S compared with control group of rats, the study parameters of plasma hemostasis did not change.

## 3. Discussion

Cerebral ischemia has several forms, depending on the nature and severity of cerebrovascular insufficiency. A number of experimental models was developed, in which the blood flow is interrupted sharply or slowly, in a whole or in a part, focally or globally, as well as permanently or transiently [27]. The models of cerebral ischemia are divided into two groups based on the extent of brain involved in ischemia: focal cerebral ischemia (FCI) and GCI [28].

Updated STAIR recommendations demand compliance with various requirements of preclinical studies for promising neuroprotectors [24]. In compliance with these requirements, proof of the efficacy of new neuroprotectors have to be obtained in no less than two species of animals. Besides, some authors suggest the necessity to study promising neuroprotectors in different models of cerebral ischemia [29,30,31]. In the opinion of Papadakis et al. [32], models of GCI cause selective neuronal death in the particular regions of the hippocampus and can be used to isolate pathways involved in the neuronal vulnerability or survival. GCI rapidly causes a cascade of events in the vulnerable regions of the brain, including the hippocampus, striatum, cerebral cortex, and cerebellum, that eventually lead to neuronal damage and death. Damage to the affected tissue can often leads to permanent impairment of mental and physical faculties, the shutdown of essential functions, or even death [33,34].

In our study, the 3VO model of GCI [35] was chosen for the experiments for the following reasons. Since this model certainly reproduces neuronal damage in CA1 region of the hippocampus, it allows one to obtain morphology data necessary to assess the possible neuroprotective activity of IQ-1S. Additionally, in the reproduction of this model, clear presentation of neurological deficit occurs [35,36], allowing one to investigate the potential mechanisms of IQ-1S action and the IQ-1S effects on neurological status. This model has been previously approved for the evaluation of neuroprotective effects of various compounds [37]. Citicoline, a neuroprotector with pleiotropic action, serves as a comparison drug. The efficacy of citicoline in acute ischemic stroke may be due to stabilization of the cell membranes, attenuation of glutamate excitotoxicity and oxidative stress, suppression of apoptosis, and elimination of endothelial dysfunction [38,39,40].

Reproduction of GCI model resulted in death of 25% of animals as well as in the development of severe symptoms of brain damage in survived animals persisting for up to five days. Slight spontaneous reduction of the severity of neurological abnormalities occurred to day 5 after reperfusion. Neuroprotective activity of IQ-1S was confirmed by studying the dynamics of neurological status in the postischemic period in animals with GCI. In case of administration of IQ-1S, the surviving animals had less severe neurological deficits in the early reperfusion period and the recovery of their neurological status was promoted. In surviving animals administered with IQ-1S and in the positive control group with administration of citicoline, the improvements of neurological status recovery were similar. We observed a decrease in mean score of neurological deficit relative to the control group. At day 1, we documented a significant decrease in the number of animals with severe neurological disorders with a corresponding increase in the number of animals with moderate degree of neurological deficit. Death rates were lower both in IQ-1S- and in citicoline-treated rats compared with control group, but these differences did not reach statistical significance. Our study confirms earlier data on the neuroprotective activity of IQ-1S in the FCI model in mice [20]. In these experiments, IQ-1S-treated mice demonstrated markedly reduced neurological deficit and infarct volumes as compared with vehicle-treated mice after 2 days of reperfusion.

Morphology data in the model of GCI confirm neuroprotective action of IQ-1S. In the settings of the GCI model used in our experiments, complete cessation of blood flow occurs in all parts of the brain without residual and collateral circulation [35]. During the 7-min episode of ischemia followed by reperfusion, diffuse damage to neural tissue occurs with predominant death of the neurons in the parts of the brain sensitive to ischemia, including pyramidal neurons of CA1 and CA3 areas in the hippocampus, small neurons of striatum and basal ganglia, and neurons in the layers II, III, and V of the neocortex [35,36]. Based on the analysis of neuroprotective activity of IQ-1S in rats with GCI, the CA1 hippocampal area was chosen as the neurons in this zone are most vulnerable to harmful effects of ischemia/reperfusion [41,42]. In the CA1 hippocampal region in the animals administered with IQ-1S, we observed a clear increase in numerical density of neurons, an increase in the number of unaltered neurons, and a significant decrease in the number of neurons with irreversible morphological damage such as pericellular edema suggesting a pronounced neuroprotective effect of the studied compound. Therefore, IQ-1S in the model of GCI demonstrated a significant neuroprotective activity, which manifested in positive effects on neurological status and morphological parameters in the CA1 area of the hippocampus.

The significance of the neuroprotective potential of IQ-1S was clearly demonstrated compared with citicoline whose neuroprotective effects have been proven experimentally [38,39,40] and in clinical trials [43,44]. Studying the neurological deficit, morphology, and lipid peroxidation products in the brain tissue did not show significant differences in the effects of IQ-1S and citicoline on these parameters despite citicoline being used in i.p. dose exceeding that of IQ-1S by 10 times. Therefore, the effects of IQ-1S and citicoline were comparable.

To exert direct neuroprotective action, the substance must penetrate the blood–brain barrier (BBB) in concentrations sufficient for therapeutic effect. Parameters of the transport through the BBB are known for the compounds, which are potential neuroprotectors [45]. We [20] calculated the following parameters of IQ-1S molecule to determine the possibility of the penetration through the BBB: the octanol/water distribution coefficient (aLogP), polar surface area of molecule (tPSA), and the number of rotating bonds (N_rot_). These calculations allowed us to determine the good ability of IQ-1S molecules to pass through the BBB.

Based on the analysis of possible direct neuroprotective/cytoprotective effect of IQ-1S, the ability of the substance to inhibit JNKs should be considered first of all [9]. JNKs are involved in many neuropathological signaling events and play a key role in the regulation of survival of brain tissues in health and disease [46]. The JNK signaling pathway plays a critical role in mediation of apoptosis in ischemia and reperfusion of the brain [34]. Numerous works demonstrate the increases in JNK phosphorylation and JNK-dependent signaling pathway activity after global and focal ischemia of the brain in rats and mice [13,20,47,48,49,50,51]. In stroke, activation of JNK aggravates brain damage, triggering inflammation and contributing to ischemic death of the cells [52]. An important role of JNK in the mechanisms of apoptosis of neuronal cells is confirmed by the experiments on knockout mice: permanent occlusion of the middle cerebral artery significantly expands the infarcted area in jnk1-/- knockout mice [4]. Further studies are needed to examine the antiapoptotic effect of IQ-1S in models of ischemia/reperfusion.

Currently, various nonprotein synthetic inhibitors of enzymatic JNK activity are described (SP600125, AS601245, IQ-1S, SR-3306, and SU3327) as well as protein and nonprotein molecules suppressing interaction between JNKs and their substrates, JNK and folding proteins and/or cellular organelles [7,8,9,10,11,12,13]. Some of them showed neuroprotective action in the models of stroke in animals [14,15,16,17]. IQ-1S is one of the most specific nonpeptide JNK inhibitors known as it does not inhibit other kinases [19]. Therapeutic efficacy of this JNK inhibitor was studied in mice with experimental cerebral ischemia and reperfusion. Introduction of IQ-1S decreased neurological deficit and infarct volume compared with those in control animals after 2 days of reperfusion [20]. Results obtained in this work using the model of acute total transitory cerebral ischemia confirmed the efficacy of IQ-1S as a neuroprotective agent.

Perhaps, the other mechanism contributing to neuroprotective effect of IQ-1S is the ability of the molecule to be an exogenous NO donor. Data showed that IQ-1S undergoes enzymatic metabolism by the liver microsomes with a release of NO confirmed by an increase in the nitrate/nitrite content in blood of the mice after i.p. administration of this compound [20]. The JNK signaling pathway is tightly coupled with NO production in ischemia and reperfusion [21]. Donors of exogenous NO attenuate S-nitrosylation of MLK3 triggered by reperfusion and inhibit the activation of the JNK-dependent pathway [22]. Indeed, SNP decreases JNK3 phosphorylation and neuronal damage in the hippocampus after global ischemia/reperfusion [53]. The involvement of other kinases into the NO-dependent modulation of JNK activity is possible. For example, S-nitrosylation of apoptosis signal-regulating kinase (ASK1) with involvement of endogenous NO activates the JNK-dependent kinase cascade during cerebral ischemia and reperfusion. At the same time, exogenous NO generated by NO donors reverses the effects of endogenous NO through the suppression of S-nitrosylation of ASK1 and exerts neuroprotection during ischemia/reperfusion [54]. Therefore, exogenous NO donors exert protective effects on the neurons through the regulation of the activities of MLK and ASK1. Other studies also suggest an important role of ASK1 in the activation of JNK-dependent pathway in ischemia/reperfusion of the brain [55].

A therapeutic approach based on the use of neuroprotectors with antioxidant properties is well founded by the concepts of pathogenesis of acute cerebrovascular disorders. Consistent with this idea, many of the world’s leading laboratories have received evidence confirming the neuroprotective effects of the antioxidant agents in cerebral ischemia [36,56]. At the same time, the presence of antioxidant activity is necessary, but insufficient for a molecule to be considered a neuroprotector. Clinical trials of individual drugs with free radical scavenging activity failed [57,58,59]. Authors who analyzed failure of the clinical trials studying the antioxidants with expected neuroprotective properties believe that the compounds of this class are promising only in the case when the antioxidant exerts multitarget neuroprotective activity and easily crosses the BBB. Our study allows us to assume that the antioxidant properties of IQ-1S may underlie its neuroprotective effects in the model of GCI. Indeed, GCI in rats was accompanied by clear activation of the lipid peroxidation, which manifested as an accumulation of primary LPPs, diene, and triene conjugates. We found that administration of IQ-1S significantly limited the LPPs accumulation in the brain tissue in the postischemic period, which could also contribute to the neuroprotective effect. This observation agrees with data showing the development of oxidative stress in patients with abnormal cerebral circulation [60].

Antioxidant properties of IQ-1S to a certain degree may be mediated by the antiradical activity of the compound. In the cells of the living organisms, the free radical processes may occur both in aqueous phase [61], and in the lipid phase of the cell membranes [62,63]. To elucidate potential antioxidant effects of the compounds using DPPH radical scavenging assay, several polar solvents with various solvating and electron-donating power may be used, including protic amphoteric solvent ethanol and aprotic bipolar solvents acetone, ethyl acetate, and acetonitrile [64]. Acetonitrile is used as a relatively nonpolar solvent. In our study, scavenging of free radicals by IQ-1S had the highest rate in ethyl acetate and acetonitrile. IQ-1S was by 1.7 times more active than ionol in acetonitrile, but IQ-1S was significantly inferior to ionol in the settings of the reaction with DPPH in regard to SPLET mechanism in ethanol. Therefore, in aprotic solvents, IQ-1S met or exceeded the antiradical activity of the standard antioxidant ionol. At the same time, the nature of antiradical activity of IQ-1S still remains unclear and this question requires further study. Considering the antioxidant effects of IQ-1S in the GCI model in rats and its antiradical properties, we suggest that the compound could decrease oxidative stress in the brain issue in ischemia/reoxygenation. Direct cytoprotective effects of IQ-1S due to JNK inhibition may be complemented with indirect antioxidant effects and effects associated with improvement of the circulation in the brain tissue.

The postischemic period is characterized by the development of hypoperfusion, which is mediated by multiple mechanisms including no-reflow phenomenon, which consists of the obstruction of the downstream microvascular bed after reperfusion of previously occluded arteries. Postischemic hypoperfusion is attributed to extrinsic compression from edema, endothelial swelling, and intravascular obstruction due to local activation of leukocytes, platelets, and coagulation [34]. In the present 3VO model of GCI, a severe postischemic hypoperfusion state occurs in the cortex of the brain [35]. Data obtained in the present study suggest that the hypoperfusion in the cerebral cortex of rats in this GCI model has steady character and persists up to five days. During this time of the postischemic period, LCBF values in the cerebral cortex in control rats were by more than twice lower compared with the corresponding values in the sham-operated animals. In rats administered with IQ-1S, the LCBF values were by 1.5 times higher than in control animals, suggesting the ability of the compound to alleviate abnormalities of circulation in the cerebral cortex.

Blood supply of the organ depends on the perfusion pressure, blood viscosity, and the vascular tone [65]. Perfusion pressure is characterized by the difference between systemic systolic blood pressure and venous pressure; i.e., it is mostly determined by arterial blood pressure. In our study, no significant differences between the mean values of arterial blood pressure in the experimental groups were observed, which probably rules out impact of the changes in the perfusion pressure on the effects of IQ-1S on LCBF. In search for other mechanisms of LCBF increase, the effects of IQ-1S on the rheological properties of blood and the tone of the isolated carotid artery in vitro were studied.

In the experiments on the endothelium-denuded rings of the carotid artery, IQ-1S exerted a dose-dependent vasodilatory effect at the concentrations ranging from 10^−9^ to 10^−4^ M. Pharmacokinetic studies of IQ-1S, performed as preclinical studies, showed that during a period from 1 to 8 h after i.g. introduction of IQ-1S at a dose of 50 mg/kg, the concentrations of the parent compound in blood plasma were in the range from 5 to 30 nM/L [66], which corresponds to the concentration range used in the experiments in vitro, where IQ-1S decreased the tone of the carotid arterial rings preconstricted by phenylephrine. Therefore, one may assume that IQ-1S can decrease tone of the cerebral blood vessels at concentrations, present in blood plasma in case of i.g. administration of IQ-1S at the used dose, and this decrease can contribute to the observed increase in LCBF in rats after GCI. Obtained data agree with previously described vasodilatory effects of other JNK inhibitor SP600125. For example, JNK activation is involved in the constriction of rat aortic smooth muscle triggered by norepinephrine, and this effect is inhibited by SP600125 [67]. In addition, SP600125 also attenuates dexmedetomidine-evoked contraction of rat thoracic aortic rings without endothelium in a concentration-dependent manner [68].

Abnormal blood rheology could be involved in the pathogenesis of microcirculation disorders and LCBF decrease in the rat cerebral cortex in the postischemic period. The presence of high blood viscosity syndrome in patients with acute abnormalities of cerebral circulation is well known [69,70]. In our study, a significant increase in blood viscosity within the entire diapason of shear rates was observed in control rats at day 5 after GCI. In rats after GCI, high blood viscosity was formed mostly due to increases in macrorheological parameters, such as hematocrit and plasma viscosity. Recovery of blood rheological properties with drugs capable to prevent or significantly attenuate the rheological abnormalities is important, but nowhere near exhausted reserve for therapy of ischemic disorders of cerebral circulation [71,72]. In our experiments IQ-1S attenuated the extent of rheological blood abnormalities in rats with GCI. In particular, blood viscosity within the entire range of shear rates was significantly lower in control and was close to the corresponding values in the group of sham-operated animals. However, IQ-1S did not significantly affect such microrheological parameters as aggregation and deformability of the erythrocytes.

Considering the risk of transformation of ischemic stroke into the hemorrhagic one when neuroprotectors are used together with the agents for thrombolytic therapy [73], we studied the parameters of plasma hemostasis in rats of experimental and control groups after GCI. IQ-1S did not show any significant effect on either of study parameters of hemostasis, which may suggest the minimal risks of its administration when there is danger of hemorrhagic transformation.

Endothelial damage may be an essential element in the pathogenesis of postischemic hypoperfusion [33]. Endothelial injury reduces the release of NO and prostacyclin, and may induce endothelin-1 production. These processes lead to increased vascular tone which may further impair blood flow in the area of the cerebral infarction and collateral vessels, thereby enhancing the ischemic injury. Endothelin-1 is a highly potent vasoconstrictor [74] to which cerebral microvessels show marked sensitivity [75]. Plasma levels of endothelin-1 are elevated in ischemic stroke [76] and are associated with cerebral edema [77]. Using the model of GCI, we demonstrated the presence of endothelial dysfunction phenomenon, which manifested as a decrease in endothelium-dependent decrease in systolic blood pressure in response to acetylcholine administration. Moreover, IQ-1S attenuated the extent of endothelial dysfunction. We assume that a decrease in blood viscosity as well as a reduction of the endothelial dysfunction in response to IQ-1S all together may result in improved microcirculation, which is seen as an increase in LCBF in the parietal cortex, i.e., an improvement of blood supply in the setting of postischemic hypoperfusion.

## 4. Materials and Methods

### 4.1. Animals

The study was carried out in accordance with the EU Directive 2010/63/EU concerning the protection of animals used for scientific purposes and approved by the Animal Care and Use Committee of Goldberg Research Institute of Pharmacology and Regenerative Medicine, Tomsk NRMC (protocol No 130092017 from 08.09.17). Experiments were carried out on 142 adult male Wistar rats (250–280 g) obtained from the Department of Experimental Biological Models of E.D. Goldberg Institute of Pharmacology and Regenerative Medicine. Rats were housed in groups of five animals per cage (57 cm × 36 cm × 20 cm) in standard laboratory conditions (ambient temperature of 22 ± 2 °C, relative humidity of 60%, light/dark period 12/12 h a day) in cages with sawdust bedding and provided with standard rodent feed (PK-120-1, Ltd., Laboratorsnab, Russia), and ad libitum water access.

### 4.2. Equipment

Rodent ventilator model 7125 (UGO Basile, Gemonio, Italy), temperature control unit HB 101/2 (Spain), homeothermic blanket control unit (Harvard Apparatus, Holliston, MA, USA), MP150 high-speed data acquisition system with matching amplifiers (Biopac Systems, Inc., Goleta, CA, USA), rotational viscometer (LVDV-II+ Pro, CP40, Brookfield Engineering Labs Inc., Middleboro, MA, USA), centrifuge model PC-6, (Dastan, Kyrgyzstan), RheoScanAnD-300 (RheoMeditech Inc., Republic Korea), rotary microtome HM340E (MICROM International GmbH, Walldorf, Germany), automatic tissue processing machine STP-120 (MICROM International GmbH, Walldorf, Germany), microscope Carl Zeiss Axio Lab.A1 and camera Axio Cam ERC5S (Carl Zeiss Microscopy GmbH, Jena, Germany), spectrophotometer Cary 50 (Varian, USA Australia), CO_2_ euthanasia device (Open Science, Russia), and Myobath II (Burghart GmbH, Wedel, Germany), coagulometer KG-4 (Cormay, Łomianki, Poland).

### 4.3. Chemicals and Drugs

The following chemicals and drugs were used in the work: Propofol-Lipuro (B. Braun Melsungen AG, Melsungen, Germany), diethyl ether for anesthesia (Kuzbassorghim, Kemerovo, Russia), Tween 80 and sodium nitroprusside dihydrate (Merck, Darmstadt, Germany), 10% neutral formalin and embedding medium Histomix (BioVitrum, Saint-Petersburg, Russia), hematoxylin (Fluca, St. Louis, MO, USA), eosin,2,3,5-triphenyl tetrazolium chloride, and ionol (Sigma, St. Louis, MO, USA), (BioVitrum, Saint-Petersburg, Russia), ethanol (Konstanta-Farm M, Moscow, Russia), acetonitrile for chromatography, acetone for chromatography, and ethyl acetate (Component-Reactive, Moscow, Russia), citicoline (Recognan, Alfa Wassermann S.p.A., Bologna, Italy), dimethyl sulfoxide (DMSO), ethylenediaminetetraacetic acid (EDTA), DPPH, acetylcholine, and phenylephrine (Sigma-Aldrich, St. Louis, MO, USA), heptane, and isopropanol (Komponent–Reaktiv, Moscow, Russia).

### 4.4. Study Molecule

Sodium salt of 11*H*-indeno[1,2-*b*]quinoxalin-11-one oxime (IQ-1S) was synthesized at the Department of Biotechnology and Organic Chemistry of Tomsk Polytechnic University, Tomsk, Russia (batch M314). Chemical structure was confirmed by the methods of mass spectrometry and nuclear magnetic resonance. Purity of the sample was 99.9%. The animals were administered with IQ-1S in the form of drug formulation. Drug formulation was the powder of the following composition: active substance of IQ-1S (28.54%) and the additive agents: carboxymethylcellulose (70.35%) and Tween 80 (1.01%). Dose of 50 mg/kg was chosen taking into account data of a dose response curve of IQ-1S effects on neurological scores and ischemic damage in the model of focal cerebral ischemia in rats as well as data of pharmacokinetic investigation (data are not shown).

### 4.5. Model of GCI

The model of GCI was reproduced by temporary blocking of blood flow in three great vessels supplying the brain, namely: the brachiocephalic artery, the left subclavian artery, and the left common carotid artery [34]. To manage general anesthesia with propofol (10 mg/kg/h), animals were implanted with a catheter placed in the right femoral vein under brief anesthesia with ethyl ether. After that, tracheal intubation was performed without damaging the integrity of the trachea. During the surgical procedure, animals were breathing spontaneously. To ligate the left common carotid artery, an access from the ventral surface of the neck was used. To ligate the brachiocephalic artery and the left subclavian artery, the surgical field was access from the first intercostal space was used. To model GCI, blood flow in the ligated blood vessels was stopped for 7 min. At the moment of spontaneous breathing secession (after 1 to 1.5 min), the intubation tube was attached to the artificial lung ventilation apparatus with an open contour. Blood flow was restored by release of the ligatures. The layered closure of the wound was performed. Upon restoring rhythmical spontaneous breathing, animals were extubated and returned to their cages with free access to food and water. Sham-operated animals were subject to similar surgical procedure, but without secession of blood flow through the ligated blood vessels.

During surgical procedure and during the entire experiment, rectal temperature in the animals was maintained at 37 ± 0.5 °C by using Homeothermic Blanket Control Unit.

### 4.6. Experimental Protocol

Animals were assigned to four groups: sham-operated (I, *n* = 23), control group (II, *n* = 46/36: total number/survived animals), experimental group (III, IQ-1S, n = 46/39), group of positive control (IV, citicoline, *n* = 20/18). Rats of group III received 50 mg/kg IQ-1S i.g. in 2 mL suspension, rats of group IV received peritoneal injection of 500 mg/kg citicoline. IQ-1S and citicoline were administrated 30 min prior to modeling GCI, and then once daily for four consequent days. Rats of control and sham-operated groups received i.g. 2 mL of physiological saline solution containing equal Tween 80 according to the same scheme. Survival and neurological status were assessed at days 1, 3, and 5 after modeling GCI. Survival was assessed in all animals taken in the experiment (135). Neurological deficit evaluation, histology study, and antioxidant activity measurement were examined in eight sham-operated rats, 16/13 rats of control group, 16/14 rats of experimental group, 14/12 rats of group of positive control. To obtain histological data, animals were randomly selected from sham-operated (*n* = 5), the control (*n* = 5), and the experimental (*n* = 5) groups at day 5 after modeling GCI. The animals were anesthetized with ethyl ether and the brains were sampled. Registration of LCBF, arterial blood pressure and heart rate were determined in five sham-operated rats, nine rats of the control group, and 10 rats of the experimental group. Measurement of hemorheological parameters was examined in five sham-operated rats, six rats of the control group, and six rats of the experimental group. Measurement of vasodilator activity of endothelium was determined in five sham-operated rats, eight rats of the control group, nine rats of the experimental group, and six rats of the positive control group. Measurement of isolated carotid artery tonus in vitro was carried out in seven intact rats. The IQ-1S effects on PT, aPTT, TT, and Fbg were studied in five sham-operated rats, eight rats of control group, and nine rats of the experimental group.

### 4.7. Neurological Deficit Evaluation

Neurological status of animals was examined by experimenter unaware of group the animals were assigned to. Neurological deficit in GCI model was evaluated based on McGraw stroke index scale [78] in our modification. The following parameters were assessed to evaluate neurological status: (1) spontaneous motor activity (normal—0 point, attenuated motor activity or stiffness, shakiness of gait, bradykinesis—1 points, elevated or absent motor activity or disorientation—2 points); (2) tail flick reflex (normal—0 point, attenuated—1 point, absent—2 points); (3) right front limb withdrawal reflexes (normal—0 point, attenuated—1 point, absent—2 points); (4) left front limb withdrawal reflexes (normal—0 point, attenuated—1 point, absent—2 points); (5) right rear limb withdrawal reflexes (normal—0 point, attenuated—1 point, absent—2 points); (6) left rear limb withdrawal reflexes(normal—0 point, attenuated—1 point, absent—2 points); (7) response to sound (normal—0 point, attenuated—1 points, absent or elevated—2 points); (8) tremor, seizures (absence—0 point, or presence—2 points); (9) muscle tones of the trunk and limbs (normal—0 point, attenuated—1 point, elevated or absent—2 points); (10) signs of ptosis (absent—0 point, unilateral—1 point, bilateral—2 points). Neurological deficit was characterized by sum of scores from all parameters. Besides, rates of animals with severe (score 6 and more), moderate (score 3–5), and mild (score less than 2) were determined in each group.

### 4.8. Histology Study

To study morphology of hippocampal neurons, rat brain was fixed in 10% neutral buffered formalin. After fixation, frontal brain segment, corresponding to coordinates from 2.80 mm to 4.30 mm relative to bregma according to rat brain stereotaxic atlas [79], were embedded in paraffin according to the commonly accepted method, sliced with rotary microtome HM 340E into 5 μm slices, and stained with hematoxylin and eosin. Biological material was processed with automatic STP 120 Spin Tissue Processor (Thermo Fisher Scientific, Waltham, MA, USA). Photodocumenting was performed with Carl Zeiss Axio Lab.A1 microscope and Axio Cam ERC5S camera. Histoquantitative study consisted of determining neuronal density by calculating number of neurons per 1 mm^2^ area of the slice and determining percentage of morphologically unchanged neurons in accordance with existing assessment criteria: clearly defined nucleus of elliptical or round shape; well identifiable nucleoli at the center of the nucleus; nucleus slightly darker than surrounding neuropil; and cytoplasm of a neuron is clearly demarcated from the surrounding neuropil. Neurons with irreversible ischemic/reperfusion injury were identified by morphological changes such as karyopyknosis, eosinophilic cytoplasm, shrunken cell body and pericellular edema [80].

### 4.9. Antioxidant Activity Measurement

To determine the lipid peroxidation products in brain tissue at day 5 after GCI, rats under ether anesthesia were decapitated; the brain was removed; one hemisphere was sampled; ethylenediaminetetraacetic acid was added to the sample at a rate of 1 mg/g of the tissue; and the sample was homogenized. Then, lipids were extracted from 250 mg of homogenate of the brain tissue by using the mixture of heptane-isopropanol (1:1) at a ratio of 1:20 (by mass) for 15 min. Phase disengagement was performed with 0.1 N HCl. The top (heptane) phase was sampled to determine the content of diene conjugates and triene conjugates. The contents of diene conjugates and triene conjugates were assessed based on absorption peaks at 232 and 275 nm characteristic of double and conjugated double bonds, respectively [81]. The content of diene and triene conjugates was determined on a Cary Win UV spectrophotometer at 232 nm and 275 nm, and expressed as OD_232_/mg lipids and OD_270_/mg lipids, correspondingly. Heptane was used as comparison solution.

Quantitative assessment of lipid content in hexane phase was done gravimetrically.

### 4.10. Antiradical Activity Measurement

Antiradical activity of IQ-1S was assessed in vitro in the model reaction of interaction with DPPH, a stable chromogen radical [82]. In the visible spectrum, DPPH in organic solvents has maximum absorption at 515–520 nm, which progressively decreases and disappears with the conversion of the compound to non-radical form on the interaction with antioxidants. Kinetic curves expressed as optical density variation for control and test samples were recorded using a spectrometer Cary 50 immediately after mixing DPPH (0.25 mM) with a solvent (control sample) or solution of test compound (0.25 mM) at volumetric ratio of 1:1. Ethanol, acetone, ethyl acetate, and acetonitrile were used as solvents. For each solvent, the experiments were performed three to six times. Antioxidant 3,5-dibutyl-4-hydroxytoluene (also known asionol, dibunole, or butylatedhydroxytoluene) was used as a comparison agent. For quantitative characteristics of antiradical activity, reaction rate of DPPH with test compounds was calculated based on the initial kinetic curve slope [83]. According to the method, antiradical activity strongly correlates with the antioxidant activity determined by this assay and can be used for quantitative assessment of antioxidant properties [83].

### 4.11. Registration of LCBF, Arterial Blood Pressure, and Heart Rate

In a designated series of experiments, the effects of IQ-1SonLCBF, heart rate, and mean arterial blood pressure in anaesthetized rats were studied on day 5 after GCI. For induction of anesthesia, diethyl ether was administered. General anesthesia was maintained with sodium thiopental i.v. through the catheter in femoral vein (at a dose of 25 mg/kg/h).

LCBF was assessed by laser-Doppler flowmetry within the primary visual cortex V1M (primary visual cortex, monocular area). The level of LCBF was registered using a data acquisition system MP150 with LDF100C module and a TSD144 fiber-optic needle-type surface sensor (25 mm × 1 mm (d)). The skin was moved apart with a retractor, and a hole with a diameter of 1.5 mm was drilled into the skull without breaking the dura mater integrity. The sensor for measuring the LCBF was installed using a special balancer-holder tool.

Mean arterial blood pressure was registered using a data acquisition system MP150 with MPMS200 and probe TSD280 femoral artery. Heart rate was calculated based on the curve of blood pressure changes. The obtained data were processed with “AcqKnowledge 4.3” software (Biopac Systems, Inc., Goleta, CA, USA).

### 4.12. Measurement of Hemorheological Parameters

Blood was sampled from the catheterized right common carotid artery and stabilized with a 1% aqueous solution of K_2_EDTA (20 μL of solution per 1 mL of blood) in Teflon-coated tubes for immediate analysis. Whole blood and plasma viscosities were measured using a rotational viscometer LVDV-II+ Pro, CP40 at 36 °C at shear rates ranging from 10.5 to 450 s^−1^. The plasma viscosity was measured at a shear rate of 450 s^−1^. The hematocrit was measured using the gravimetric method by centrifuging blood samples in glass capillaries at 1300 g for 20 min; the results are expressed as a percentage. Red blood cell (RBC) aggregation and deformability were measured with RheoScanAnD-300. The critical time of aggregation (T_1/2_) (Ct) and the critical stress (Cs) of aggregation were used to characterize the RBC aggregation. The elongation index (EI) was used to characterize RBC deformability [84,85].

### 4.13. Measurement of Vasodilator Activity of Endothelium

To evaluate functional state of the vascular endothelium, animals were implanted with a catheter in the right femoral artery for registration of arterial blood pressure and a catheter in the right femoral vein for bolus introduction of pharmacological agents. Arterial blood pressure was restarted using a data acquisition system MP150 with MPMS200 and TSD280 sensor. The obtained data were processed with “AcqKnowledge 4.3” software. Hypotensive reaction of systemic arterial blood pressure in response to endothelium-dependent vasodilation and endothelium-independent vasodilation were registered according to the method of Laursen et al. [86] in our modification. Endothelium-dependent vasodilation mediated by NO-synthase activation was triggered by bolus i.v. administration of Ach at a dose of 40 μg/kg; area above the curve of blood pressure decrease S_Ach_ was calculated and expressed in mm Hg•sec. Endothelium-independent vasodilation was triggered by i.v. bolus introduction of SNP at a dose of 30 μg/kg with calculation of S_SNP_ by the same approach. To quantitatively characterize vasodilatory activity of the vascular endothelium, the IVA was used, calculated as ratio of S_Ach_/S_SNP_.

### 4.14. Measurement of Isolated Carotid Artery Tonus In Vitro

Wistar rats were euthanized by asphyxiation in a CO_2_ chamber. Carotid arteries were dissected, cleaned of adherent fat, and de-endotelized with means of 3-mm rings of carotid arteries were prepared and placed in an organ chamber of Myobath II between two stainless hooks for isometric recording. One hook was attached to the bottom of the bath, and the other was connected with the isometric force transducer. Rings were stretched to a resting tension of 1 g and equilibrated for 1 h. In all experiments, carotid artery rings were preconstricted with 30-µM phenylephrine (PE) after the tone of the vascular preparations was stabilized. IQ-1S was studied at concentrations ranging from 0.001 to 100 μM. Test substance was diluted in 100% DMSO to achieve the concentration of 0.01 M. Further concentrations were obtained by subsequent dilution in 100% DMSO. Final concentration of DMSO in solution was 1% and did not significantly affect contractile activity of vascular rings.

### 4.15. Measurement of the Pasma Hemostasis Parameters

Prothrombin time (PT), activated partial thromboplastin time (aPTT), thrombin time (TT), and fibrinogen level (Fgn) were identified to assess the state of plasma hemostasis in platelet-poor plasma. The blood was centrifuged to receive platelet poor plasma at centrifugation speed of 1300 g for 20 min. The study of plasma hemostasis parameters was performed using the KG-4 coagulometer with test assays “PT-test” (prothrombin time), “Techplastin-test” (activated partial thromboplastin time), “Thrombo-test” (thrombin time), “Fibrinogen-test” (fibrinogen) (“Technology-Standard”, LLC, Russia).

### 4.16. Statistical Analysis

Statistical processing was performed with Statistica 8.0 software. All results are expressed as mean ± SEM. Group variation was assessed with Kruskal–Wallis test. Significant difference between were assessed by Fisher’s exact test for severity of neurological deficit, by Mann–Whitney U-test for average rate of neurological deficit and infarct volume, by Student’s *t*-test for histological and antiradical investigation. Compliance of the sample with normal distribution was evaluated with Shapiro–Wilk’s W test and Kolmogorov–Smirnov & Lilliefors test. Values were considered statistically significant when *p* was < 0.05.

## 5. Conclusions

IQ-1S at a dose of 50 mg/kg demonstrated pronounced neuroprotective properties in the model of GCI when administered i.g. in a treatment-and-prophylactic regime. Neuroprotective effect was observed as a significant decrease in the loss of the pyramid neurons in the CA1 hippocampal area highly sensitive to ischemia as well as a clearer reduction of neurological deficit after GCI episode. Regarding the neuroprotective effect, IQ-1S met or exceeded citicoline used as a positive control. We suggest that the beneficial effects of IQ-1S on stroke outcome could be a combined result of JNK inhibition by the parent compound in brain tissue, NO generation during its bioconversion, their antioxidant and antiradical activity, improvement of microcirculation in cerebral tissue as result of vasodilatory activity, and decrease of blood viscosity and endothelial dysfunction. Our results suggest that IQ-1S may be considered as a neuroprotector with a multitarget mechanism of action.

## Figures and Tables

**Figure 1 molecules-24-01722-f001:**
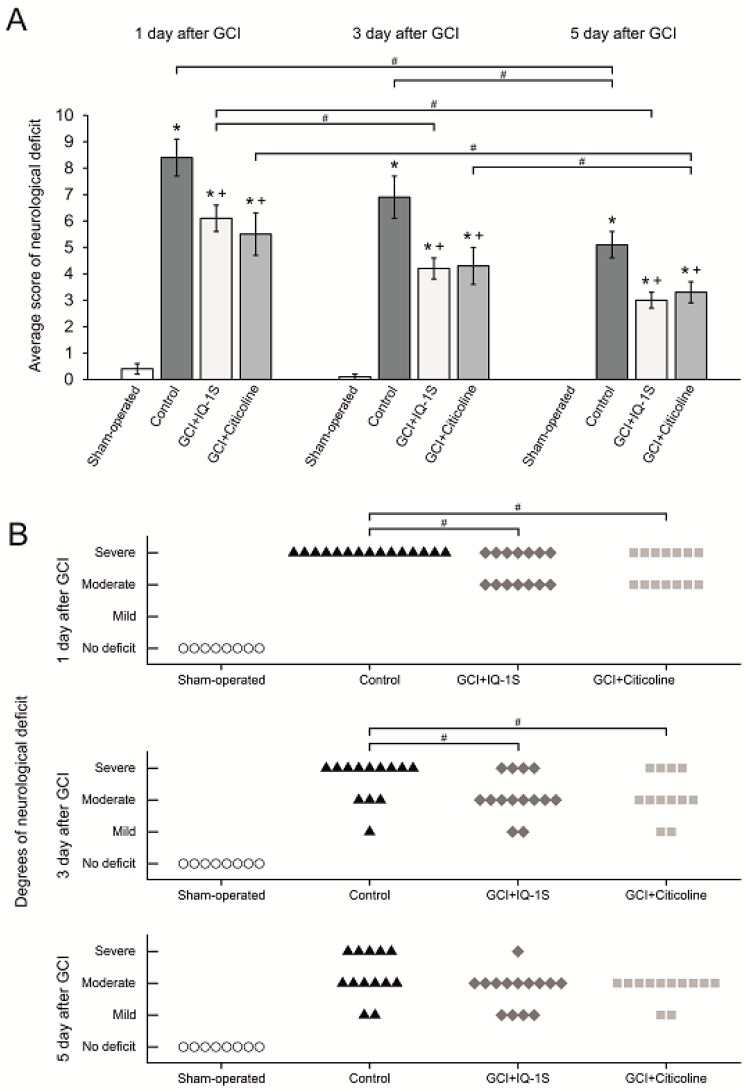
Effects of IQ-1S (50 mg/kg, i.g.) and citicoline (500 mg/kg, i.p.) on average score of neurological deficit (**A**) and animal distribution for degree of neurological deficit (severe, moderate, and mild) (**B**). * *p* < 0.05 as compared with the sham-operated animals; + *p* < 0.05 as compared with the control animals (A: # *p* < 0.05 at different times; B: # *p* < 0.05 for subgroups “severe” only).

**Figure 2 molecules-24-01722-f002:**
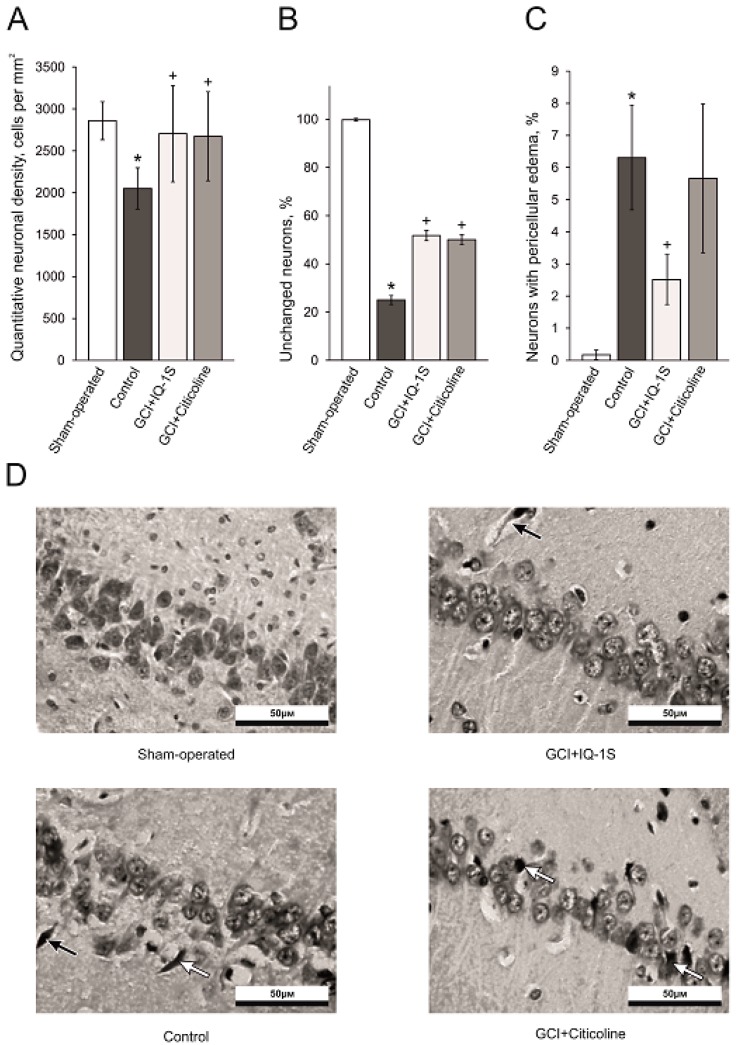
Effects of IQ-1S (50 mg/kg, i.g.) and citicoline (500 mg/kg, i.p.) on the quantitative neuronal density (**A**), percentage of unchanged neurons (**B**), percentage of neurons with pericellular edema (**C**) and illustrations of CA1-zone of hippocampus (**D**) in rats at day 5 after GCI. * *p* < 0.05 as compared with the sham-operated animals; ^+^
*p* < 0.05 as compared with the control. Black arrows indicate neurons with pericellular edema; white arrows indicate pyknotic eosinophilic neurons.

**Figure 3 molecules-24-01722-f003:**
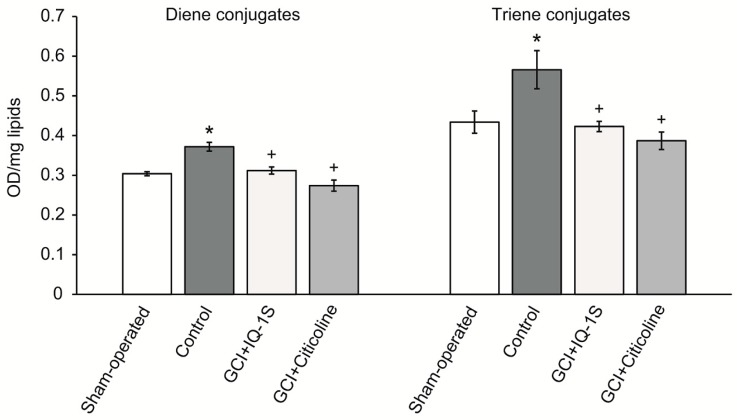
Effects of IQ-1S (50 mg/kg, i.g.) on contents of diene conjugates (OD_232_/mg of lipids) and triene conjugates (OD_270_/mg of lipids) in the brain tissue in rats at day 5 after GCI. * *p* < 0.05 as compared with the sham-operated animals; ^+^
*p* < 0.05 as compared with the control.

**Figure 4 molecules-24-01722-f004:**
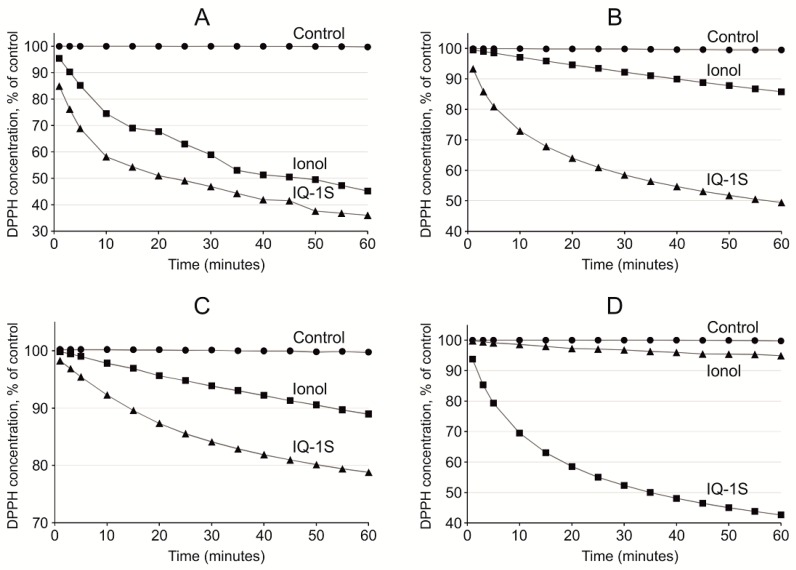
Decay of DPPH radical in control sample and during the reaction with IQ-1S and ionol in acetone (**A**), in acetonitrile (**B**), in ethyl acetate (**C**), and in ethanol (**D**). The initial concentrations of DPPH radical and IQ-1S or ionol were always the same.

**Figure 5 molecules-24-01722-f005:**
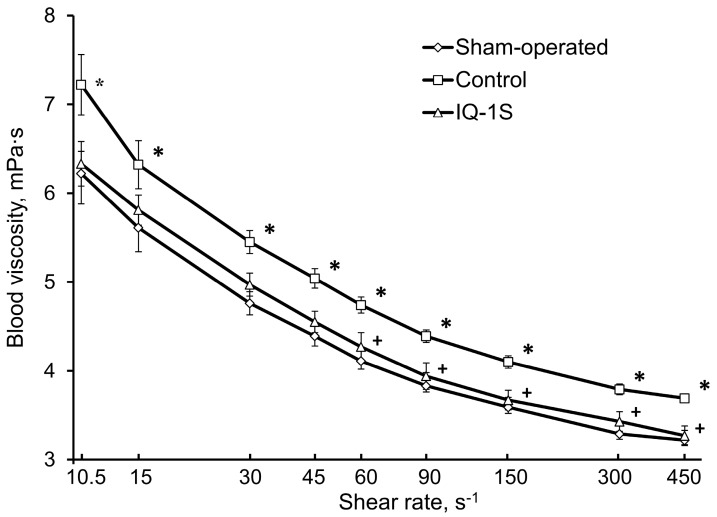
Effects of IQ-1S (50 mg/kg, i.g.) on the viscosity of whole blood (mPa·s) in rats at day 5 after GCI. * *p* < 0.05 as compared with the sham-operated animals; ^+^
*p* < 0.05 as compared with the control.

**Figure 6 molecules-24-01722-f006:**
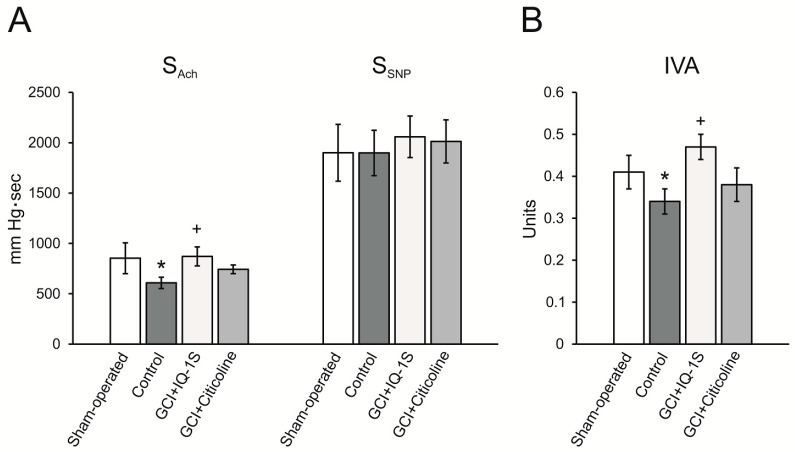
Effects of IQ-1S (50 mg/kg, i.g.) and citicoline (500 mg/kg, i.p.) on the endothelium-dependent vasodilation, induced by acetylcholine (S_Ach_) and SNP (S_SNP_) (**A**), and index of vasodilatory activity (IVA = S_Ach_/S_SNP_) (**B**) in rats at day 5 after GCI. * *p* < 0.05 as compared with the sham-operated animals; ^+^
*p* < 0.05 as compared with the control.

**Figure 7 molecules-24-01722-f007:**
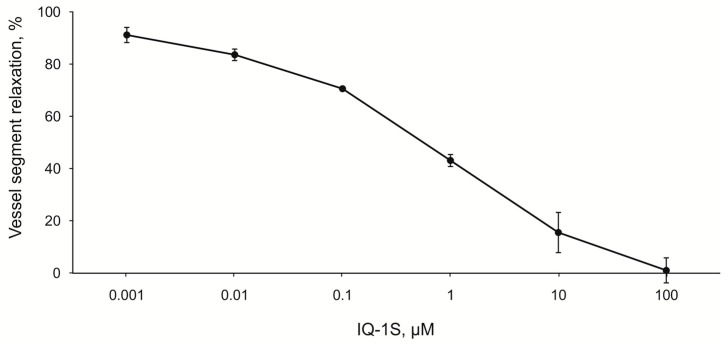
The effects of IQ-1S on the tone of the vascular rings, isolated from the carotid artery and constricted by phenylephrine (1 μM).

**Table 1 molecules-24-01722-t001:** The effects of IQ-1S and citicoline on mortality of animals after global cerebral ischemia (GCI).

Group of Animals	Number of Dead Animals (% of Total Number)		
	Day 1	Day 3	Day 5
Sham-operated (*n* = 23)	0 (0)	0 (0)	0 (0)
Control (*n* = 46)	4 (9)	9 (20)	10 (22)
IQ-1S, 50 mg/kg (*n* = 46)	5 (11)	7 (15)	7 (15)
Citicoline, 500 mg/kg (*n* = 20)	0 (0)	2 (10)	2 (10)

**Table 2 molecules-24-01722-t002:** Reaction rate constants (min^−1^) DPPH with IQ-1S and ionol in various solvents.

Solvent	IQ-1S	Ionol
Ethyl acetate	0.0802 ± 0.0148	0.0448 ± 0.0126
Acetonitrile	0.0395 ± 0.0028 ^+^	0.0031 ± 0.0005
Acetone	0.0123 ± 0.0013 ^+^	0.0017 ± 0.0008
Ethanol	0.0021 ± 0.0011 ^+^	0.0456 ± 0.0013

^+^*p* < 0.05 as compared with ionol.

**Table 3 molecules-24-01722-t003:** The effects of IQ-1S (50 mg/kg) on LCBF (BPU, a blood perfusion unit), mean arterial blood pressure (ABP, mm Hg), and heart rate (HR, min^−1^) in rats at day 5 after GCI.

Group	LCBF	ABP	HR
Sham-operated (*n* = 5)	1900 ± 191	93 ± 9	358 ± 10
Control (*n* = 9)	888 ± 80 *	97 ± 2	376 ± 13
IQ-1S (*n* = 10)	1424 ± 213 ^+^	94 ± 4	352 ± 12

* *p* < 0.05 as compared with the corresponding values in sham-operated rats; ^+^
*p* < 0.05 as compared with the corresponding values in control rats.

**Table 4 molecules-24-01722-t004:** The effects of IQ-1S (50 mg/kg, i.g.) on plasma viscosity, hematocrit (Ht), and parameters of erythrocyte aggregation in the shear flow (Ct and Cs) in rats at day 5 after GCI.

Group	Plasma Viscosity, mPa∙s	Ht, %	Ct, s	Cs, mPa
Sham-operated (*n* = 5)	1.16 ± 0.01	42 ± 1	9.3 ± 0.3	144.1 ± 4.2
Control (*n* = 6)	1.19 ± 0.01 *	45 ± 1 *	10.6 ± 0.7	145.6 ± 11.3
IQ-1S (*n* = 6)	1.17 ± 0.01	42 ± 1 ^+^	9.6 ± 0.2	151.8 ± 4.6

* *p* < 0.05 as compared with the corresponding values in sham-operated rats; ^+^
*p* < 0.05 as compared with the corresponding values in control rats.

**Table 5 molecules-24-01722-t005:** The effects of IQ-1S (50 mg/kg, i.g.) on the erythrocyte elongation index in rats at day 5 after GCI.

Group	Shear Stress, Pa				
	1	3	7	10	20
Sham-operated (*n* = 5)	0.201 ± 0.007	0.354 ± 0.005	0.453 ± 0.004	0.486 ± 0.003	0.533 ± 0.002
Control (*n* = 6)	0.192 ± 0.006	0.348 ± 0.005	0.449 ± 0.004	0.483 ± 0.003	0.532 ± 0.003
IQ-1S (*n* = 6)	0.198 ± 0.006	0.356 ± 0.005	0.455 ± 0.003	0.487 ± 0.002	0.532 ± 0.003

**Table 6 molecules-24-01722-t006:** The effects of IQ-1S (50 mg/kg, i.g.) on fibrinogen (Fbg), the activated partial thromboplastin time (aPTT), prothrombin time (PT), and thrombin time (TT) in rats at day 5 after GCI.

Group	Fbg, g/L	aPTT, s	PT, s	TT, s
Sham-operated (*n* = 5)	2.68 ± 0.28	17.8 ± 0.7	25.3 ± 0.5	32.8 ± 0.4
Control (*n* = 9)	3.30 ± 0.21	17.9 ± 0.5	26.3 ± 0.6	24.5 ± 3.9
IQ-1S (*n* = 10)	3.04 ± 0.14	17.3 ± 0.5	26.3 ± 0.3	27.9 ± 3.0

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
