# Peer review of "Protective Effects of a New C-Jun N-terminal Kinase Inhibitor in the Model of Global Cerebral Ischemia in Rats"

_molecules, 2019, doi:10.3390/molecules24091722_

Round 1

Reviewer 1 Report

This investigation looked at neuroprotective properties of JNK inhibitors. JNK inhibitors have shown neuroprotective potential in various stroke models and present a viable strategy for stroke treatment. The idea of developing JNK inhibitors is an interesting. I have following comments for authors;

Artificial ventilator for rats were not used during the surgery. Without it, animals can go into respiratory depression and skew the outcomes.

The anesthesia used in the study is not appropriate for the type of study.

There was no use of rCBF measurement without which it is hard to ascertain the occlusion of arteries and reduction of CBF.

LPP estimated in brain tissues but not in hippocampus.

Global ischemia is a model of delayed hippocampal neuronal death which takes 10-12 days to show the effect. In this study,only 5 day after GCI was utilized.

Citicoline has failed in clinical trials, not sure why was it used as a positive control. 

NDS used in the methods is 4-point deficit method, however in the figure 1 A, the scale bar shows 10-points.

Author Response

Response to Reviewer 1

Dear Reviewer,

Thank you for your time and efforts to review our manuscript! We appreciate your valuable comments and suggestions. Please find below our responses:

> This investigation looked at neuroprotective properties of JNK inhibitors. JNK inhibitors have shown neuroprotective potential in various stroke models and present a viable strategy for stroke treatment. The idea of developing JNK inhibitors is an interesting. I have following comments for authors; Artificial ventilator for rats were not used during the surgery. Without it, animals can go into respiratory depression and skew the outcomes.

Reply: Surgical intervention in modeling of GCI was performed in the presence of preserved spontaneous rhythmic breathing in animals. Surgical procedure with access to ligated vessels through the first intercostal space ruled out a possibility of the development of pneumothorax whereas the used level of propofol-based narcotization preserved spontaneous rhythmic breathing. The use of artificial ventilation would prevent using the registration of the fact of a spontaneous breathing cessation as a sign of cerebral ischemia in GCI modeling.

> The anesthesia used in the study is not appropriate for the type of study.

Reply:

(1) Propofol is on the list of the recommended anesthetics for conducting the studies on the models of cerebral ischemia in rats [Percie du Sert et al. 2017].

(2) While developing the GCI model, the comparative study was performed to compare chloral hydrate often used as an agent for general anesthesia for modeling of cerebral ischemia in rats [Manual of Stroke Models in Rats, 2006] with propofol in regard to survival of rats, average score for neurological deficit, and animal distribution for degree of neurological deficit. Data showed that these parameters did not significantly differ both when chloral hydrate and propofol were used [Atochin et al. 2017].

Ref.:

1. Percie du Sert N., Alfieri A, Allan SM, Carswell HV, Deuchar GA, Farr TD, Flecknell P, Gallagher L, Gibson CL, Haley MJ, Macleod MR, McColl BW, McCabe C, Morancho A, Moon LD, O'Neill MJ, Pérez de Puig I, Planas A, Ragan CI, Rosell A, Roy LA, Ryder KO, Simats A, Sena ES, Sutherland BA, Tricklebank MD, Trueman RC, Whitfield L, Wong R, Macrae IM. The IMPROVE Guidelines (Ischaemia Models: Procedural Refinements Of in Vivo Experiments). J Cereb Blood Flow Metab. 2017;37(11):3488-3517.

2. Manual of Stroke Models in Rats. Y. Wang-Fischer (ed.), 1st Edition, CRC Press, 2008; Handbook of Experimental Neurology: Methods and Techniques in Animal Research, T. Tatlisumak, M. Fisher (eds.), Cambridge University Press, 2006.

3. Atochin DN, Chernysheva GA, Aliev OI, Smolyakova VI, Osipenko AN, Logvinov SV, Zhdankina AA, Plotnikova TM, Plotnikov MB. An improved three-vessel occlusion model of global cerebral ischemia in rats. Brain Res. Bull. 2017;132: 213–221.

> There was no use of rCBF measurement without which it is hard to ascertain the occlusion of arteries and reduction of CBF.

Reply:

In this study, we did not perform the registration of rCBF for assessment of ischemia degree because we tried to avoid additional surgical stress and disruption of anatomical continuity of the skull. The following considerations allow to avoid the necessity of the registration of rCBF:

1) Researchers involved in this work are the authors of this model [Atochin et al. 2017. Patent RU 2544369]. Unlike 4VO models, in this 3VO model, high reproducibility of total ischemia has been shown, which was demonstrated in the registration of local CBF and the use of staining [Atochin DN. et al., 2017]. The laboratory has a sufficient experience in implementing this model in other experiments [Atochin et al. 2016, Anishchenko et al. 2018, Chenysheva et al. 2018], which allows to make sure of its reproducibility.

2) In this GCI model, occlusion of the left common carotid artery, the brachiocephalic artery, and the left subclavian artery is done under direct vision. In response to the occlusion, definite GCI signs appear: during the first minute, respiratory arrest occurs; visible part of choroid changes color from red to dark-gray with the following powerful mydriasis and the absence of pupillary reflex to a bright light.

Ref.:

1. Atochin DN, Chernysheva GA, Aliev OI, Smolyakova VI, Osipenko AN, Logvinov SV, Zhdankina AA, Plotnikova TM, Plotnikov MB. An improved three-vessel occlusion model of global cerebral ischemia in rats. Brain Res. Bull. 2017;132: 213–221.

2. Patent RU 2544369: http://www1.fips.ru/fips_servl/fips_servlet?DB=RUPAT&rn=6117&DocNumber=2544369&TypeFile=html.

3. Atochin DN, Chernysheva GA, Smolyakova VI, Osipenko AN, Logvinov SV, Zhdankina AA, Sysolyatin SV, Kryukov YA, Anfinogenova Y, Plotnikova TM, Plotnikov MB. Neuroprotective effects of p-tyrosol after the global cerebral ischemia in rats. Phytomedicine. 2016;23(7):784–792.

4. Anishchenko AM, Aliev OI, Sidekhmenova AV, Shamanaev AYu, Kutchin AV, Chukicheva IYu, Torlopov MA, Plotnikov MB. The neuroprotective effect of d-HES on total transient cerebral ischemia in rats. Bulletin of Experimental Biology and Medicine. 2018;165(6):728–730.

5. Chenysheva GA, Smol'akova VA, Kutchin AV, Chukicheva IY, Plotnikov M.B. Neuroprotective effects of dibornol in focal cerebral ischemia/reperfusion in rats. Bulletin of Experimental Biology and Medicine. 2018;166(1):15–18.

> LPP estimated in brain tissues but not in hippocampus.

Reply: You are right. Unfortunately, we did not find the description of micro-method for research of the content of diene conjugates and triene conjugates and used macro-method with estimation of LPP in total brain.

> Global ischemia is a model of delayed hippocampal neuronal death which takes 10-12 days to show the effect. In this study, only 5 day after GCI was utilized.

Reply: According to opinion of M. Fisher et al. [Fisher et al. 2009], search for promising neuroprotectors must involve integrated assessment of neurological deficit and data of morphological study of the brain tissue. It means that, in the absence of expected differences in one of these parameters (namely: neurological deficit) between control and experiment, the significance of studying neuroprotective activity is lost. In the model of global ischemia of the brain in rats after three days, the processes of recovery actively occur and the effects of neurological deficit diminish. To illustrate this, we present the tables with the results of average score of neurological deficit and animal distribution for degree of neurological deficit (severe, moderate, and mild) for 7 days, obtained in the earlier performed studies showing that at day 7 after GCI, severe degree was not observed in any of animals whereas mild degree was present in majority of animals.

Average score of neurological deficit

Group

After GCI

1 day

3 day

5 day

7 day

Scam-operated (n=5)

0.4±0.3

0

0

0

Control (n=11)

8.6±1.2

6.3±1.4

4.7±0.5

3.0±0.5

Animal distribution for degree of neurological deficit (% of rats in group)

Degree of neurological deficit

After GCI

1 day

3 day

5 day

7 day

Control (n=11)

Severe (score > 6)

77.8

25

14.3

0

Moderate (score 3–5)

22.2

75

85.7

42.9

Mild (score < 2)

0

0

0

57.1

We would like also to note that there were no significant differences between rats of control group and rats administered with IQ-1S and citicoline as soon as at day 5 (see manuscript, Fig. 1B, day 5 after GCI).

Involution of the neurological deficit in rats to days 3-5 after GCI has been described by other authors:

1. Johansson SE, Larsen SS, Povlsen GK, Edvinsson L. Early MEK1/2 inhibition after global cerebral ischemia in rats reduces brain damage and improves outcome by preventing delayed vasoconstrictor receptor upregulation. PLoS One. 2014;9(3):e92417.

2. Wang P, Yao L, Zhou LL, Liu YS, Chen MD, Wu HD, Chang RM, Li Y, Zhou MG, Fang XS, Yu T, Jiang LY, Huang ZT. Carbon monoxide improves neurologic outcomes by mitochondrial biogenesis after global cerebral ischemia induced by cardiac arrest in rats. Int J Biol Sci. 2016;12(8):1000-9.

In this regard, we limited duration of our study by the period of five days.

Ref.:

Fisher M, Feuerstein G, Howells DW, Hurn PD, Kent TA, Savitz SI, Lo EH, STAIR Group. Update of the stroke therapy academic industry roundtable preclinical recommendations. Stroke, 2009;40:2244–2250.

> Citicoline has failed in clinical trials, not sure why was it used as a positive control. 

Reply:

Citicoline treatment within 14 days after stroke onset improves the outcome in patients with acute ischemic stroke, as compared with placebo, which is confirmed by meta-analysis of 10 randomized clinical studies:

1. Secades JJ, Alvarez-Sabín J, Castillo J, Díez-Tejedor E, Martínez-Vila E, Ríos J, Oudovenko N. Citicoline for acute ischemic stroke: A systematic review and formal meta-analysis of randomized, double-blind, and placebo-controlled trials. J. Stroke Cerebrovasc. Dis. 2016;25(8):1984-1996.

2. Dávalos A, Alvarez-Sabín J, Castillo J, Díez-Tejedor E, Ferro J, Martínez-Vila E, Serena J, Segura T, Cruz VT, Masjuan J, Cobo E, Secades JJ.; International Citicoline Trial on acUte Stroke (ICTUS) trial investigators. Citicoline in the treatment of acute ischaemic stroke: an international, randomised, multicentre, placebo-controlled study (ICTUS trial). Lancet. 2012;380(9839):349–357.

Citicoline is used as a positive control for research and development of potential neuroprotectors:

1. Yuliani S, Mustofa, Partadiredja G. The neuroprotective effects of an ethanolic turmeric (Curcuma longa L.) extract against trimethyltin-induced oxidative stress in rats. Nutr. Neurosci. 2018, Mar 7. 1–8.

2. Wang NQ, Wang LY, Zhao HP, Liu P, Wang RL, Song JX, Gao L, Ji XM, Luo YM. Luoyutong Treatment Promotes Functional Recovery and Neuronal Plasticity after Cerebral Ischemia-Reperfusion Injury in Rats. Evid. Based Complement. Alternat. Med. 2015;2015. Article ID 369021. 12 p.

3. Bustamante A, Giralt D, Garcia-Bonilla L, Campos M, Rosell A, Montaner J. Citicoline in pre-clinical animal models of stroke: a meta-analysis shows the optimal neuroprotective profile and the missing steps for jumping into a stroke clinical trial. J. Neurochem. 2012, 123(2):217–225.

> NDS used in the methods is 4-point deficit method, however in the figure 1 A, the scale bar shows 10-points.

Reply:

Indeed, a degree of neurological abnormalities in FCI model was determined by the method based on 4-point deficit method: the tail suspension test, posture maintenance, circling test, and horizontal righting reflex [Zhang et al. 2015].

Neurological deficit in GCI model was evaluated based on McGraw stroke index scale [McGraw et al. 1977]. The following parameters were assessed to evaluate neurological status:

1) spontaneous motor activity (normal – 0 point, attenuated motor activity or stiffness, shakiness of gait, bradykinesis – 1 points, elevated or absent motor activity or disorientation – 2 points);  

2) tail flick reflex (normal – 0 point, attenuated – 1 point, absent – 2 points);

3) right front limb withdrawal reflexes (normal – 0 point, attenuated – 1 point, absent – 2 points);

4) left front limb withdrawal reflexes (normal – 0 point, attenuated – 1 point, absent– 2 points);

5) right rear limb withdrawal reflexes (normal – 0 point, attenuated – 1 point, absent– 2 points);

6) left rear limb withdrawal reflexes(normal – 0 point, attenuated – 1 point, absent– 2 points);

7) response to sound (normal – 0 point, attenuated – 1 points, absent or elevated – 2 points);  

8) tremor, seizures (absence – 0 point, or presence – 2 points);

9) muscle tones of the trunk and limbs (normal– 0 point, attenuated – 1 point, elevated or absent– 2 points);

10) signs of ptosis (absent– 0 point, unilateral– 1 point, bilateral– 2 points).

In the revised version of the manuscript, detailed quantitative characteristics of all parameters used for the assessment of neurological deficit are given. Besides, the clarification was added to the text: Neurological deficit was characterized by sum of scores from all parameters.

Maximum value of neurological deficit in our experiments was 8.5 (GCI, control, day 1). Therefore, the 10-point scale bar was chosen for convenience.

Ref.:

Zhang H, Shen Y, Wang W, Gao H. Rat model of focal cerebral ischemia in the dominant hemisphere. Int. J. Clin. Exp. Med. 2015;8(1): 504–511.

McGraw CP. Experimental cerebral infarction effects of pentobarbital in Mongolian gerbils. Arch. Neurol. 1977; 34:334–336.

Reviewer 2 Report

The manuscript by Plotnikov et al describes the protective effects of IQ-1S, a new c-jun N-terminal kinase inhibitor, in the model of global cerebral ischemia in rats. In particular, the Authors performed experiments showing that IQ-1S-induced neuroprotection is due to its pleiotropic effects. Although the topic of the study is not extremely innovative, and the authors already published two papers on the neuroprotective effects of IQ-1s in cerebral ischemia, the experimental design of the present paper is well organized and the experiments well conducted. However, some aspects of the study need to be clarified and some additional experiments are necessary to completely address the working hypothesis.

Major points

1.    A dose response curve of IQ-1S effects on neurological scores and ischemic damage should be performed.

2.    The neuroprotective effect of IQ-1S should be observed for a time lag longer than five days. These experiments could provide more relevance to the beneficial effects of IQ-1S on stroke outcomes.

3.    In figure 1. Panel A as far as concern the neurological scores, the authors should underline if a statistically significant difference occurs in rats subjected to GCI at different times (1, 3 and 5 days). Moreover, an explanation about the criteria used to classify the severity of neurological deficit has to be provided (panel B).   

4.    As far as concern the effect of IQ-1S on neuronal density and on the percentage of the unchanged neurons in hippocampus the authors should provide evidence of the molecular mechanisms involved. Is IQ-1S able to stimulate neurogenesis in hippocampus or it is able to counteract apoptosis? Experiments should be performed to address these questions.

5.    The serum levels of NO should be detected in rats exposed to GCI and treated with IQ-1S as well as to citicoline.

6.    The effects of IQ-1S on PT and aPTT should be explored in order to demonstrate that the treatment with IQ-1S does not expose rats to risk of bleeding.Considered the effects of IQ-1S on vasodilatator activity of endothelium, experiments addressed to evaluate the presence of cerebral edema should be performed in rats treated with IQ-1S

7.    To give more emphasis to the neuroprotective effects of IQ-1S further experiments might be performed in rats subjected to tMCAO. In this model the effects of IQ-1S on reoxygenation could be assessed. These experiments will allow to further confirm  the antioxidant properties of  IQ-1S

8.    Although the pleiotropic effects of IQ-1S, the amount of neuroprotection observed is similar in rats treated with IQ-1S or with citicoline. This aspect of the study should be clarified in the discussion in a more consistent way

Minor points

1.    In each figure should be explained that the group of animals exposed to GCI is considered the control. Please indicate GCI as control

2.    Discussion is too long and should be shortened

Author Response

Response to Reviewer 2

Dear Reviewer,

Thank you for your time and efforts to review our manuscript! We appreciate your valuable comments and suggestions. Please find below our responses:

> The manuscript by Plotnikov et al describes the protective effects of IQ-1S, a new c-jun N-terminal kinase inhibitor, in the model of global cerebral ischemia in rats. In particular, the Authors performed experiments showing that IQ-1S-induced neuroprotection is due to its pleiotropic effects. Although the topic of the study is not extremely innovative, and the authors already published two papers on the neuroprotective effects of IQ-1s in cerebral ischemia, the experimental design of the present paper is well organized and the experiments well conducted. However, some aspects of the study need to be clarified and some additional experiments are necessary to completely address the working hypothesis.

> Major points

> 1. A dose response curve of IQ-1S effects on neurological scores and ischemic damage should be performed.

Reply:

1. A dose response curve of IQ-1S effects on neurological scores and ischemic damage was studied in the model of FCI. The results of this investigation were described in the manuscript "Neuroprotective effects of a new inhibitor of c-Jun N-terminal kinases in the rat model of transient focal cerebral ischemia", which is under consideration in Drug Research Development journal (ID is DDR-19-0020). Based on the results described in the above-mentioned article and data of pharmacokinetics, we chose IQ-1S dose of 50 mg/kg, which was studied. This information was added to the manuscript (4.4. Study Molecule).

2. There are the journals whose Author Requirements specify that the study must be carried out with the assessment of dose/effect [for example, Phytomedicine]. There is no such a requirement in Molecules journal.

> 2.  The neuroprotective effect of IQ-1S should be observed for a time lag longer than five days. These experiments could provide more relevance to the beneficial effects of IQ-1S on stroke outcomes.

Reply:

The task of this investigation was to decide on the neuroprotective activity of IQ-1S in the model of transient global cerebral ischemia in rats and to find out its mechanisms (including pleiotropic). We deliberately limited the study by duration of five days

1) In the model of global ischemia of the brain in rats after three days, the processes of spontaneous recovery are actively ongoing and the signs of neurological deficit fade. To illustrate this, we provide the tables on Average score of neurological deficit and Animal distribution for degree of neurological deficit (severe, moderate, and mild) during 7 days. At day 7 after GCI, no animals had severe degree whereas most animals had mild degree.

Average score of neurological deficit

Group

After GCI

1 day

3 day

5 day

7 day

Scam-operated (n=5)

0.4±0.3

0

0

0

Control (n=11)

8.6±1.2

6.3±1.4

4.7±0.5

3.0±0.5

Animal distribution for degree of neurological deficit (% of rats in group)

Degree of neurological deficit

After GCI

1 day

3 day

5 day

7 day

Control (n=11)

Severe (score > 6)

77.8

25

14.3

0

Moderate (score 3–5)

22.2

75

85.7

42.9

Mild (score < 2)

0

0

0

57.1

We would like also to note that as soon as at day 5, the significant differences between rats of control group and rats administered with IQW-1S or citicoline were absent (see manuscript, Fig. 1 B, day 5 after GCI).

Involution of the neurological deficit in rats to days 3-5 after GCI has been described by other authors:

1. Johansson SE, Larsen SS, Povlsen GK, Edvinsson L. Early MEK1/2 inhibition after global cerebral ischemia in rats reduces brain damage and improves outcome by preventing delayed vasoconstrictor receptor upregulation. PLoS One. 2014;9(3):e92417.

2. Wang P, Yao L, Zhou LL, Liu YS, Chen MD, Wu HD, Chang RM, Li Y, Zhou MG, Fang XS, Yu T, Jiang LY, Huang ZT. Carbon monoxide improves neurologic outcomes by mitochondrial biogenesis after global cerebral ischemia induced by cardiac arrest in rats. Int J Biol Sci. 2016;12(8):1000-9.

In this regard, we limited duration of our study by the period of five days.

According to opinion of M. Fisher et al. [Fisher et al. 2009], the evaluation of neurological deficit data and data of morphological study of brain tissue is the most important as these parameters characterize neuroprotective activities of compounds the most. From this point of view, in the absence of differences in one of these parameters (neurological deficit) between control and experiment, the significance of studying neuroprotective activity is lost. In this regard, we limited duration of our study by the period of five days.

Ref.:

Fisher M, Feuerstein G, Howells DW, Hurn PD, Kent TA, Savitz SI, Lo EH, STAIR Group. Update of the stroke therapy academic industry roundtable preclinical recommendations. Stroke, 2009;40:2244–2250.

> 3. In figure 1. Panel A as far as concern the neurological scores, the authors should underline if a statistically significant difference occurs in rats subjected to GCI at different times (1, 3 and 5 days). Moreover, an explanation about the criteria used to classify the severity of neurological deficit has to be provided (panel B).

Reply:

Statistically significant differences at different times (1, 3 and 5 days) in Figure 1, Panel A is indicated. An explanation about the criteria used to classify the severity of neurological deficit is added in 4.7. Neurological Deficit Evaluation.   

> 4.    As far as concern the effect of IQ-1S on neuronal density and on the percentage of the unchanged neurons in hippocampus the authors should provide evidence of the molecular mechanisms involved. Is IQ-1S able to stimulate neurogenesis in hippocampus or it is able to counteract apoptosis? Experiments should be performed to address these questions.

Reply:

1. The study of the effects of IQ-1S on neurogenesis is an independent task we plan to compete. However, it is impossible in a framework of study design we chose to evaluate the neuroprotective activity. Five days after GCI is an insufficient time for manifestation of the effects of these compounds on neurogenesis. Earlier, in the GCI model, we studied the effects of two compounds stimulating neurogenesis: fluoxetine and p-tyrosol. The effects of these compounds on the neurogenesis manifested at days 10 and 31 of the study [Kisel et al. 2016, Khodanovich et al. 2018, Khodanovich et al. 2016].

2. We have received preliminary data suggesting that neuroblastoma NMB7 cells treated with IQ-1S (10 µM) demonstrate significantly less doxorubicin-induced apoptosis. We plan a detailed evaluation of anti-apoptotic effects of the JNK inhibitor in different experimental models, related to ischemia-reperfusion. Results of the study will be reported in separate publication together with data on JNK signaling in neurogenesis after ischemia-reperfusion. Indeed, mechanisms of neurogenesis, cell proliferation, and anti-apoptosis in the ischemic area are tightly regulated in brain [Guo et al. 2017]. In the revised version of the manuscript, we added an additional sentence in the text that further studies are needed to examine anti-apoptotic effect of our JNK inhibitor in the models of ischemia-reperfusion (see Page 12, Paragraph 4).

Ref.:

1. Kisel AA, Chernyshova GA, Smol'yakova VI, Savchenko RR, Plotnikov MB, Khodanovich MYu. Modulation of hippocampal neurogenesis in the global brain ischemia model: effects of p-tyrosol and fluoxetine. Annals of Neurology. 2016;80(Suppl 20):S101–102.

2. Khodanovich M, Kisel A, Kudabaeva M, Chernysheva G, Smolyakova V, Krutenkova E, Wasserlauf I, Plotnikov M, Yarnykh V. Effects of fluoxetine on hippocampal neurogenesis and neuroprotection in the model of global cerebral ischemia in rats. Int. J. Mol. Sci. 2018;19(1):E162.

3. Khodanovich MYu, Kisel AA, Chernyshova GA, Smol'yakova VI, Savchenko RR, Plotnikov MB. Effect of Fluoxetine on Neurogenesis in Hippocampal Dentate Gyrus after Global Transient Cerebral Ischemia in Rats. Bul. Exp. Biol. Med. 2016;161(3):351–354.

4. Guo F, Lou J, Han X, Deng Y, Huang X. Repetitive Transcranial Magnetic Stimulation Ameliorates Cognitive Impairment by Enhancing Neurogenesis and Suppressing Apoptosis in the Hippocampus in Rats with Ischemic Stroke. Front Physiol. 2017;8:559

> 5. The serum levels of NO should be detected in rats exposed to GCI and treated with IQ-1S as well as to citicoline.

Reply:

We [Schepetkin et al. 2016] demonstrated the ability of IQ-1S to exert properties of NO donor in vitro and in vivo. In our opinion, this is sufficient to discuss one of the mechanisms of possible neuroprotective activity of IQ-1S. We consider a repetition of these studies in the GCI model unnecessary. Unlike IQ-1S, structure of citicoline molecule does not have any groups able to create NO radicals. Therefore, we believe that there is no theoretical basis for conducting the identical study with citicoline.

> 6.  The effects of IQ-1S on PT and aPTT should be explored in order to demonstrate that the treatment with IQ-1S does not expose rats to risk of bleeding. Considered the effects of IQ-1S on vasodilatator activity of endothelium, experiments addressed to evaluate the presence of cerebral edema should be performed in rats treated with IQ-1S.

Reply:

1. While understanding the risk of transformation of ischemic stroke into the hemorrhagic one in performing main series of IQ-1S studies in GCI model, we concurrently studied the effects of IQ-1S on PT, aPTT, TT, and fibrinogen. The available data have been added to the manuscript (P. 11, Paragraph 1, Table 6).

2. The presence of pericellular edema in the neurons of hippocampal CA-1 zone was estimated. The available data have been added to the manuscript (P. 4: 2.2. Effects of IQ-1S on the Morphological Structure of CA1 Hippocampal Area; Figure 2).

> 7. To give more emphasis to the neuroprotective effects of IQ-1S further experiments might be performed in rats subjected to tMCAO. In this model the effects of IQ-1S on reoxygenation could be assessed. These experiments will allow to further confirm the antioxidant properties of IQ-1S

Reply:

We performed a separate study of neuroprotective activity of IQ-1S in the model of tMCAO in rats. This material has an independent significance and is presented in the manuscript "Neuroprotective effects of a new inhibitor of c-Jun N- terminal kinases in the rat model of transient focal cerebral ischemia" that has been submitted to Drug Development Research journal (ID is DDR-19-0020) for consideration.

> 8. Although the pleiotropic effects of IQ-1S, the amount of neuroprotection observed is similar in rats treated with IQ-1S or with citicoline. This aspect of the study should be clarified in the discussion in a more consistent way

Reply:

In Discussion section, the effects of IQ-1S and citicoline are compared (P. 12, Paragraph 3).

> Minor points

> 1. In each figure should be explained that the group of animals exposed to GCI is considered the control. Please indicate GCI as control

Reply: Done.

> 2. Discussion is too long and should be shortened

Reply: The Discussion section was shortened.

Round 2

Reviewer 1 Report

No comments. All my concerns are addressed

Reviewer 2 Report

The authors only partially replied to the referee requests. However, the quality of the manuscript was slightly improved.